# Leveraging Patient Reported Outcomes Measurement via the Electronic Health Record to Connect Patients with Cancer to Smoking Cessation Treatment

**DOI:** 10.3390/ijerph17145034

**Published:** 2020-07-13

**Authors:** Julia R. May, Elizabeth Klass, Kristina Davis, Timothy Pearman, Steven Rittmeyer, Sheetal Kircher, Brian Hitsman

**Affiliations:** 1Department of Preventive Medicine, Feinberg School of Medicine, Northwestern University, Chicago, IL 60611, USA; elizabeth.klass@northwestern.edu; 2Robert H. Lurie Comprehensive Cancer Center of Northwestern University, Chicago, IL 60611, USA; t-pearman@northwestern.edu (T.P.); Sheetal.kircher@nm.org (S.K.); 3Quality Innovation Center, Northwestern Medicine, Chicago, IL 60611, USA; kristina.davis@nm.org; 4Department of Medical Social Sciences, Feinberg School of Medicine, Northwestern University, Chicago, IL 60611, USA; 5Information Systems, Northwestern Medicine, Chicago, IL 60611, USA; steveritz2001@yahoo.com

**Keywords:** cancer, smoking cessation, patient reported outcomes, electronic health record, oncology, quality improvement

## Abstract

Tobacco use negatively impacts cancer treatment outcomes, yet too few providers actively support their patients in quitting. Barriers to consistently addressing tobacco use and referring to treatment include time constraints and lack of knowledge surrounding treatment options. Patient Reported Outcomes (PRO) measurement is best practice in cancer care and has potential to help address these barriers to tobacco cessation treatment. This descriptive program evaluation study reports preliminary results following implementation of a novel automated PRO tobacco use screener and referral system via the electronic health record (EHR) patient portal (MyChart) that was developed and implemented as a part of a population-based tobacco treatment program at the Robert H. Lurie Comprehensive Cancer Center of Northwestern University. Between 25 June 2019 and 6 April 2020, 4589 unique patients completed the screener and 164 (3.6%) unique patients screened positive for recent (past month) cigarette smoking. All patients who screened positive were automatically referred to a smoking cessation treatment program integrated within the Lurie Cancer Center, and 71 (49.7%) patients engaged in treatment, as defined by completing at least one behavioral counseling session. Preliminary results indicate that the PRO/MyChart system may improve smoker identification and increase offering of treatment and, despite the “cold call” following a positive screen, may result in a treatment engagement rate that is higher than rates of treatment engagement previously documented in oncology settings. Longer term evaluation with formal statistical testing is needed before drawing conclusions regarding effectiveness, but PRO measurement via the EHR patient portal may serve a potentially important role in a multi-component approach to reaching and engaging cancer patients in comprehensive tobacco cessation treatment.

## 1. Introduction

Continued smoking following a cancer diagnosis increases both cancer-specific and all-cause mortality [1]. Smoking is also associated with adverse treatment outcomes and treatment related toxicity for patients undergoing chemotherapy, radiation, or surgery [2,3,4,5]. Although quitting smoking decreases risks of mortality and may lead to improved treatment outcomes, many patients who smoked prior to their diagnosis will continue to smoke [4]. The National Comprehensive Cancer Network (NCCN) recommends that smoking status be assessed at every visit and that all smokers are referred to treatment regardless of cancer prognosis [6]. Best practice is an “opt out” method, in which all patients are screened for tobacco use and referred, regardless of willingness to quit [7]. One commonly accepted model of tobacco assessment and treatment is the “Ask, Advise, Refer” method, in which all patients are asked about tobacco use, advised to quit, and referred to evidence-based treatment [8]. Although many providers ask and advise, only 42% of providers report providing treatment or referring patients to treatment [9,10]. Barriers to consistently providing tobacco cessation support include time constraints and low confidence in ability to adequately treat patients [11]. 

Patient Reported Outcomes (PRO) measurement is best practice in cancer care and has potential to overcome clinician-related barriers to addressing tobacco use among cancer patients. PRO measurement is the method of collecting information regarding the status of a patient’s health condition, behavior, or experience directly from the patient [12]. PRO measurement is effective in oncology settings to routinely screen for distress and automatically route results to appropriate providers for treatment [13,14]. Patients with cancer exhibit a high prevalence of distress symptoms, such as fatigue, pain, and anxiety. As for smoking status, these symptoms are often unaddressed during an oncology clinic visit due to similar barriers of time constraint and feelings of inability to adequately address symptoms [15,16,17]. Automated PRO measurement of tobacco use and treatment referral via the electronic health record (EHR) patient portal has not been studied, but may be effective in the oncology setting due to the potential to address barriers to care by enabling systematic administration of screening, automated referral for patients who smoke, repeated offering of treatment to smokers not interested in treatment, and potentially to reduce stigma associated with continued tobacco use after a cancer diagnosis. 

The PRO measurement approach is designed to supplement, not replace, standard assessment and referral by clinicians, and there are several reasons why it could be valuable in an oncology setting. While clinician-initiated tobacco use screening and treatment referral is effective, it often may not be feasible given the range of urgent medical and psychological issues that require attention during an oncology visit. As mentioned previously, most clinicians report asking patients about tobacco use at the initial visit; however, only about 38% re-assess tobacco use at follow-up visits [18]. Given the high smoking relapse rate and the unique timeline of greatest relapse risk for patients with cancer, relying on provider assessment may result in a missed smoking lapse or relapse [19,20]. Clinicians vary in their knowledge of evidence-based methods for treating tobacco use and studies have shown that oncologists prefer smoking cessation treatment to be handled by another health care provider [16,21].

Among healthcare contexts in which PRO measurement is utilized, providers report greater confidence in their ability to treat patients and improved communication with patients [22,23]. Providers also report greater satisfaction with repeated screening across the disease course [24]. Willingness to disclose smoking to the oncology care team may be limited, as many patients under-report their smoking when asked by their physician [25]. PRO measurement has the benefit of greater privacy and can be administered electronically in the clinic or remotely via the EHR patient portal and has been found to be feasible and effective in most settings [26]. Consistent with an “opt-out” approach, PRO measurement via the EHR patient portal can be systematically administered at regular intervals, with referrals made regardless of patients’ readiness to quit or interest in treatment.

The Robert H. Lurie Comprehensive Cancer Center of Northwestern University/Northwestern Memorial Hospital has employed PRO measurement via the EHR patient portal since December 2014 [27]. The initial version of the PRO screener measured fatigue, physical functioning, pain, depression and anxiety, as well as practical, health education, and dietary needs. The system was based on the NIH’s Patient Reported Outcomes Measurement Information System (PROMIS) and is a computerized, adaptive test. In mid-2019, we added a 5-item questionnaire to screen cancer patients for lifetime smoking (Figure 1). The underlying motivation was to address gaps in cancer care in the Lurie Cancer Center: low rates of tobacco use assessment and documentation in the EHR system and treatment referral. In the current study, we utilized a descriptive program evaluation design to report the preliminary effectiveness of a novel PRO tobacco use screener and referral system automated via the EHR patient portal, which was developed and implemented as a part of a population-based treatment program. The study was approved by Northwestern University’s Institutional Review Board (STU00210245).

## 2. Materials and Methods

The automated PRO tobacco screener and referral system was implemented via the patient portal of the EHR system of Lurie Cancer Center on 25 June 2019. Lurie Cancer Center utilizes Epic Systems EHR. The 5-question screener (Figure 1) assesses lifetime cigarette use, frequency and intensity of current use, and degree of tobacco dependence [28]. Patients eligible for screener completion, determined by having a cancer-related ICD-10 code, receive an email 72 hours before a scheduled appointment with their medical or surgical oncologist notifying them that they have a questionnaire available in the patient portal, Epic MyChart (Figure 2). Patients enrolled in MyChart, estimated to be 83.4% of patients as of March 2020, may complete the screener via MyChart. Patients receive a bundled set of PRO measures, including pain, physical function, fatigue, depression/anxiety, and other potential adverse impacts of cancer treatment. The measures are assigned together, but due to EHR functionality, some questionnaires, including the smoking screener, appear as separate questionnaires in the patient view. 

At the end of the assessment, patients are informed that it may take up to 72 hours for the results to be evaluated, and if they are experiencing any emergencies, they should call 911 or proceed to the nearest emergency department. Patients who do not complete the screener from home or who are not enrolled in MyChart can complete the screener in the clinic while waiting to meet with their provider or via MyChart after the appointment until midnight, when the screener expires. To minimize completion burden, patients are assigned the questionnaire a maximum of once per month. EHR implementation of PRO measurement is described in the PROs Implementation Planning Toolkit [29]. 

Assessment results immediately populate in the EHR as unique questionnaire series submissions patient encounters. These results are viewable in the system by all care team members and can be added to progress notes using smart-text shortcuts. For patients who report smoking within the past 30 days, results are automatically routed to the EHR in-basket of the tobacco cessation staff pool (Figure 3).

When a positive screen message is received by the Tobacco Treatment Specialist (TTS), a standardized workflow (Figure 4; darkened boxes) is triggered to ensure prompt connection to treatment, beginning with documentation of the patient’s referral and adding the patient to the outreach queue. If the patient uses MyChart, the TTS messages the patient via the patient portal within 24 hours concerning their recently reported smoking and offers treatment. If the patient does not use MyChart or does not answer the initial MyChart message, s/he receives up to three phone calls to offer treatment. If the patient cannot be reached after three calls, s/he is marked as “unable to contact” and taken off the outreach queue unless referred again. If contact is made, a tobacco treatment staff member, trained in motivational interviewing, conducts a brief intake to confirm the patient’s eligibility for services and offers to schedule an initial consultation with a TTS. All patients with a current or past cancer diagnosis seen in the cancer center who smoke, defined as smoking even a puff within the last 30 days, are eligible for treatment. The tobacco cessation treatment program is based on the NCCN smoking cessation practice guidelines and offers all patients behavioral counseling, either in person or via telephone, as well as medications (varenicline, bupropion, nicotine replacement therapy) and referrals as needed to external resources, such as the Illinois Tobacco Quitline or the National Cancer Institute’s SmokefreeTXT. All patient outreach attempts are documented in the EHR, and a communication is sent to providers following treatment engagement or decline. Treatment updates are also provided to the oncologist.

Occurring at the same time as the launch of the MyChart automated PRO tobacco screening and referral system, a provider referral option was implemented within the EHR. Program staff conducted trainings with providers and staff throughout the departments of the Lurie Cancer Center on the importance of addressing tobacco use and how to refer to the tobacco cessation treatment program using the provider referral method. An overview of all smoker identification and treatment referral options, shown in Figure 4 (column of boxes on the far left), was also presented.

After at least three months of treatment, all patients who engaged in at least one treatment visit received a telephone call requesting completion of a satisfaction survey. The 5-question survey assessed patients’ satisfaction with their tobacco cessation treatment program experience, their prescribed smoking cessation medication (if applicable), counseling sessions with the TTS, likelihood of recommending the tobacco cessation treatment program, and their level of agreement with the statement “The Tobacco Cessation Program improved my care at Lurie Cancer Center.” Patients were asked to rate their responses on a 4-point scale, with 1 = lowest satisfaction and 4 = highest satisfaction. Two call attempts were made, and patients were given the option to decline answering specific questions or the entire questionnaire. 

After a 9-month observation period following the PRO system implementation, EHR data extraction was conducted in collaboration with analysts from the Northwestern University Enterprise Data Warehouse and the Northwestern Memorial Hospital Analytics Department. Smoking screener data were extracted utilizing an Epic report that monitors questionnaire completion status. Sociodemographic data concerning the patient population were extracted from the EHR through a custom Enterprise Data Warehouse report consisting of all departments within the cancer center. For quality assurance, data were validated in collaboration with members of Northwestern Medicine Analytics.

## 3. Results

Results are displayed in Figure 5. Between 25 June 2019 and 6 April 2020, 15,318 unique patients received the PRO tobacco screener and 4,589 (29.9%) unique patients completed it. Of completed screeners, 164 (3.6%) unique patients screened positive for current/recent smoking, with the remaining screeners (70.1%) expired at midnight on the appointment date due to incompletion. Of the 164 unique patients, 9 (5.5%) patients were ineligible due to erroneous screener assignment. One hundred forty-three (87.2%) unique patients were considered “reached” either by completion of phone outreach protocol or infusion drop in and 71 (49.7%) patients engaged in treatment. During this time period, the total number of smokers in the cancer center was estimated to be about 1529, or roughly 4.2% of all patients seen. The true prevalence may be higher due to inconsistent smoking status documentation, as only 82% of patients were assessed at least once during the observation period.

During the observation period, 59 unique patients were referred to treatment via the EHR by their oncology care team. Of the 59 unique patients, 53 (89.8%) were reached through phone outreach or in person during an infusion visit, and 36 (67.9%) patients were engaged in treatment. All patients who engage in treatment are offered in-person or telephone-based counseling and medication per NCCN guidelines. Of the 71 patients who engaged in treatment through the PRO system, 49 (69%) completed at least one in-person behavioral counseling session, and 46 (64.8%) completed at least one telephone counseling session. Sixty (84.5%) patients used a smoking cessation medication, defined as either self-reported use of a medication or receipt of a prescription through the treatment program. 

Patients who were engaged in treatment through the PRO system had a moderate to high degree of tobacco dependence: 64.3% were everyday smokers, 44.8% smoked 10 or more cigarettes per day, and 55.2% smoked within 30 minutes of waking. Engaged patients had a variety of primary tumor types, with the greatest representation from gastrointestinal (19.7%), hematological (18.4%), breast (16.9%), and lung/thoracic (15.5%). Table 1 provides details regarding the sociodemographic characteristics of all smokers in the Lurie Cancer Center patient population and those referred and engaged in treatment. 

Twenty-seven (38.6%) patients who were engaged via the PRO system and 9 (25%) patients who were engaged via provider referral answered some or all of the satisfaction survey questions. Some patients did not respond to certain questions or chose to not complete the entire survey. When asked if they were satisfied with their overall tobacco cessation program experience, 91.7% (22/24) of PRO referred patients responded “yes, completely” or “yes, for the most part”, compared to 75% (6/8) of provider referred patients. All PRO referred patients (*n* = 23) and all provider referred patients (*n* = 7) stated that they were “completely” or “for the most part” satisfied with their counseling sessions with the TTS. When asked if the tobacco cessation treatment program improved their overall care at the Lurie Cancer Center, 95.7% (22/23) of PRO referred patients stated that they “agreed” or “definitely agreed”, compared to 75% (6/8) of provider referred patients. Lastly, 95.8% (23/24) of patients stated that they would be “likely” or “very likely” to recommend the tobacco cessation treatment program to another patient at the Lurie Cancer Center who smoked, compared to 77.8% (7/9) of patients referred by their provider.

## 4. Discussion

Our experience with automated PRO/MyChart tobacco screener and referral system at Lurie Cancer Center over the first 9 months of operation suggests that it may help address common barriers to tobacco use assessment and referral to evidence-based treatment. A main advantage is the reliable assessment of tobacco use. The automatic assignment of the 5-item smoking screener that is linked with each encounter with the oncologist, up to 12 times per year, ensures that smoking status is assessed at least once monthly. By supplementing the “Ask Assess, Refer” approach in the clinic setting, the PRO system may maximize the likelihood of identifying all smokers and referring them for treatment. In practice, only 40% of providers actively assist their patients with quitting or refer them to treatment, creating a critical missed opportunity [9,10]. This gap in cancer care is being addressed by Lurie Cancer Center and 41 other cancer centers under the NCI Cancer Moonshot Cancer Center Cessation Initiative (C3I). Initial results from the C3I centers, including the current preliminary study by our center, show major gains in establishing integrated tobacco cessation programs and reaching and treating smokers with cancer [30,31,32,33,34,35,36,37]. These studies have also identified key leverage points, including workflow efficiency and leadership and stakeholder support of implementation [35] and EHR system modifications to facilitate tobacco use assessment and treatment referral and delivery [33,34,36]. Areas in need of further improvement, such as increasing reach to racial and ethnic minorities to address disparities in cancer-related burden, have also been identified [31].

Insufficient treatment referral, which has been documented in many oncology settings, was observed in the current study. Despite analytics reports estimating 1529 smokers seen in the Lurie Cancer Center between 25 June 2019 and 6 April 2020, only 59 (3.9%) patients were referred by their provider. The PRO system resulted in 164 (10.7%) referrals, nearly 3 times as many referrals as the provider pathway. Only 6 patients were referred through both the provider and PRO system pathways. It is possible that provider referrals were limited due to knowledge that the PRO system was in place; however, provider trainings were focused on the importance of provider referrals and continued monitoring and support of patients’ quit attempts and long-term smoking cessation. 

PRO measurement via the EHR patient portal may also capture patients who initially decline treatment referral offered by providers. Though we do not have data on the number of patients who declined a treatment referral offered by their provider, 2 of the 6 dual referred patients had initially declined an offer of treatment following provider referral but later engaged in treatment after outreach triggered by a PRO/MyChart positive screen and referral. This may prove advantageous to re-offering treatment services at regular intervals to patients who initially decline treatment after a cancer diagnosis.

The systematic and automated screening process may also capture slips or relapses that occur among former smokers, as was observed in the current study. Several PRO system referrals involved patients with “former smoker” status. Self-reported tobacco use is generally accurate for current smokers and never smokers with cancer, as well as patients who quit 1 year prior to diagnosis [38]. However, reporting accuracy may decrease for patients who report quitting within 1 year of diagnosis [38]. This should be taken into account for PRO measurement responses that reflect former smoker status and may warrant additional outreach to former smokers.

There is some indication that PRO measurement via the patient portal may lead to an increased overall satisfaction with cancer care, and this was observed in our study [39]. Continued smoking following a cancer diagnosis, as well as distress symptoms, are known to impact patient quality of life and treatment outcomes [40,41]. Distress in cancer patients, particularly when left unrecognized, has been shown to lead to poorer satisfaction in cancer care [42]. The tobacco cessation treatment program workflow response to a positive screen provides consistent and timely offering of treatment, which may be a contributing factor for higher patient satisfaction rates observed for PRO system referred patients compared to provider referred patients. This finding is preliminary based on the small subgroup of patients who completed the satisfaction survey. 

The preliminary success of the automated PRO tobacco screener and referral system may be due in part through reducing the stigma that many smokers experience following their cancer diagnosis, especially smoking attributable cancer [43]. This experience may lead to under-reporting of smoking in the face-to-face context with providers [44]. PRO measurement via the EHR patient portal provides an opportunity to lessen the feeling of stigma by self-reporting remotely rather than in person to the care team. 

PRO measurement may address potential implicit care-team biases in smoking assessment and referral. Our preliminary results suggest that providers referred more patients of a racial minority and those insured by Medicaid, both key disparity groups, compared to the PRO approach. It is not yet clear if the PRO system reaches a different population than the provider approach, but the consistent and standardized assessment and automated referral helps ensure that all patients are assessed for tobacco use and all smokers receive a referral to treatment. An important goal for the future refinement of the PRO system will be to identify ways to better reach patients who most need to be connected with tobacco treatment, such as patients of low socioeconomic status and racial minorities. Potential strategies include telephone screening for patients who are not enrolled in MyChart, geo-fenced targeted ad campaigns for specific underrepresented zip codes, and in-clinic completion of the PRO tobacco screener.

Brief advice to quit by healthcare provider has positive effects on a patient’s smoking cessation progress [45,46], leading to a potential criticism of the PRO measurement approach which generally operates outside of the oncology setting. It is reassuring that our initial results indicate that the PRO system approach may produce high engagement rates, though lower than those observed for provider referrals (49% versus 68%, respectively). Both approaches appear to produce higher than the average engagement in treatment, which has been found to be about 20% in oncology settings [47,48]. Optimizing reach and engagement will likely require a multicomponent approach comprised of automated PRO tobacco screening and referral via the patient portal prior to oncology visits, provider referral of patients seen in the oncology clinic, and a tobacco use registry to identify and proactively reach smokers who are not receiving cancer treatment. Continued reinforcement by oncology providers of the importance of quitting and tobacco cessation treatment for all patients who are reached could help to maximize treatment engagement. Another limitation of the PRO system is the potential difficulty for patients with low health or computer literacy or those with visual or motor limitations. It also currently only reaches patients who speak English or those with access to someone who is able to translate. A long-standing aim has been to improve PRO measurement in diverse patient populations [49].

A final limitation is that we observed a high screener non-completion rate (~70%). Demographic and smoking characteristic data are not yet available for patients who did not complete the screener. These data will be critical in guide future improvements to enable broader reach. In the meantime, training is ongoing to ensure that oncology care teams continue to actively encourage their patients to complete assigned PRO measures via the patient portal before their oncology visits and to improve efficiency of in-clinic completion when needed. Tablets in the clinics to enable patients to complete screeners while waiting for appointments may also assist with completion rates and are being considered in the Lurie Cancer Center. While we observed an estimated low smoking prevalence (3.6%), this is comparable to other studies of cancer patients of NCI-designated cancer centers [32,33]. The low smoking prevalence may also be associated with the sociodemographic characteristics of our cancer patient population (e.g., middle to high socioeconomic status). The low PRO tobacco screener completion rate (~30%) reinforces the importance of a multicomponent approach to reaching patients.

Several factors have been important to the initial success of the PRO tobacco screening and automated referral system. First, administrative and clinical leadership buy-in has been crucial for ensuring smooth and widespread implementation. Physician support of EHR-based screening is low, but can be increased when physicians are actively involved at the pre-implementation stage [23]. Engage physicians early and seek champions in multiple departments to facilitate widespread implementation and regularly remind providers of their critical role in reinforcing smoking cessation and treatment utilization. Second, regular updates to providers about referral outcomes and treatment plans and outcomes is essential. The extent to which providers believe that the PRO tobacco screening and automated referral adds value will be addressed in future research. Third, to enable reach to diverse patients in terms of cancer and cancer treatment, the PRO system should be implemented in all departments of a cancer center. This will require comprehensive training within and across departments to ensure the active support of cancer care teams, which in turn may increase screener completion rates [50]. Finally, the outreach protocol triggered by a positive screen must be efficient and timely to ensure rapid connection to treatment.

## 5. Conclusions

The automated PRO tobacco screening and referral system described in this study has the potential to overcome common cancer patient and oncology provider barriers by automating the systematic assessment and documentation of smoking status and treatment referral of all smokers, independent of readiness to quit or level of interest in treatment. Based on our experience, implementation success relies heavily on cancer center leadership buy-in, physician engagement, especially at the pre-implementation stage, and an efficient and effective workflow. Preliminary results indicate that the PRO/MyChart measurement system may improve smoker identification and increase offering of treatment and, despite the “cold call” following a positive screen, may result in a treatment engagement rate that is higher than rates of treatment engagement previously documented in oncology settings. Longer term evaluation with formal statistical testing is needed before drawing conclusions regarding effectiveness, but PRO measurement via the EHR patient portal may serve a potentially important role in a multi-component approach to reaching and engaging cancer patients in comprehensive tobacco cessation treatment.

## Figures and Tables

**Figure 1 ijerph-17-05034-f001:**
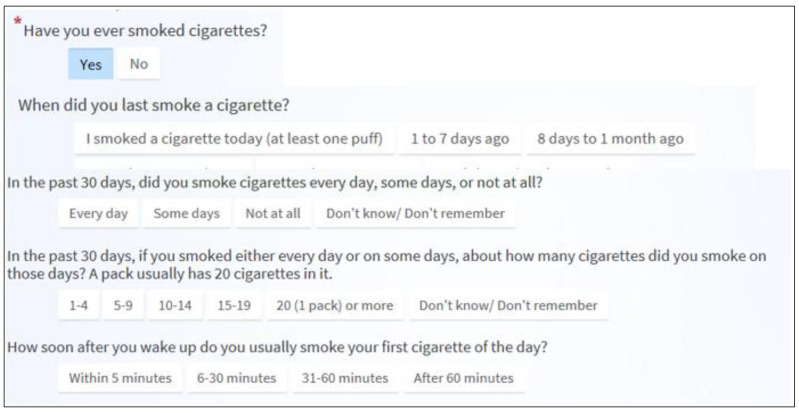
Patient view of tobacco screener items. (Nelson, 2020). © 2020 Epic Systems Corporation. Used with permission.

**Figure 2 ijerph-17-05034-f002:**
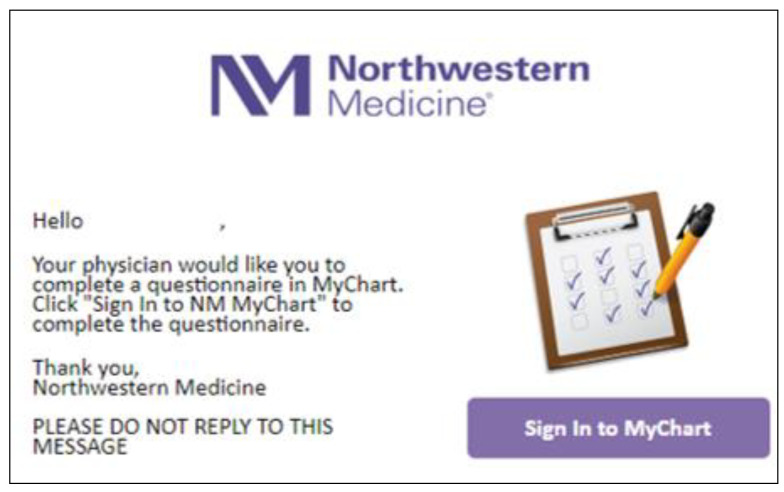
Patient e-mail notification for the tobacco screener questionnaire assignment. © 2020 Epic Systems Corporation. Used with permission.

**Figure 3 ijerph-17-05034-f003:**
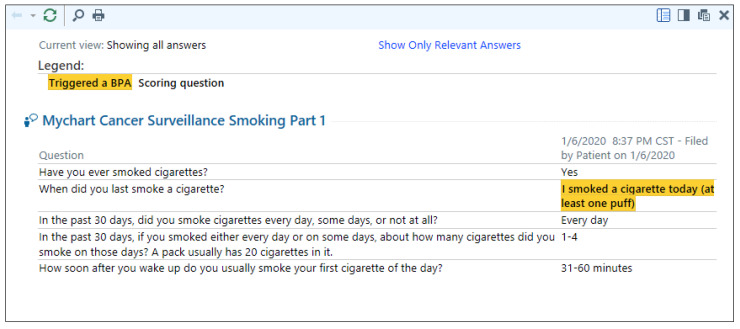
Provider and program view of a positive screen result. © 2020 Epic Systems Corporation. Used with permission.

**Figure 4 ijerph-17-05034-f004:**
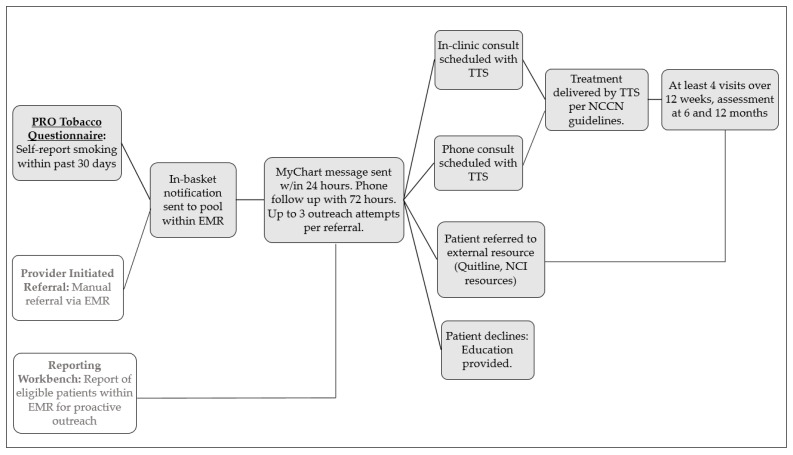
The workflow of the tobacco cessation program from smoker identification to treatment engagement and delivery.

**Figure 5 ijerph-17-05034-f005:**
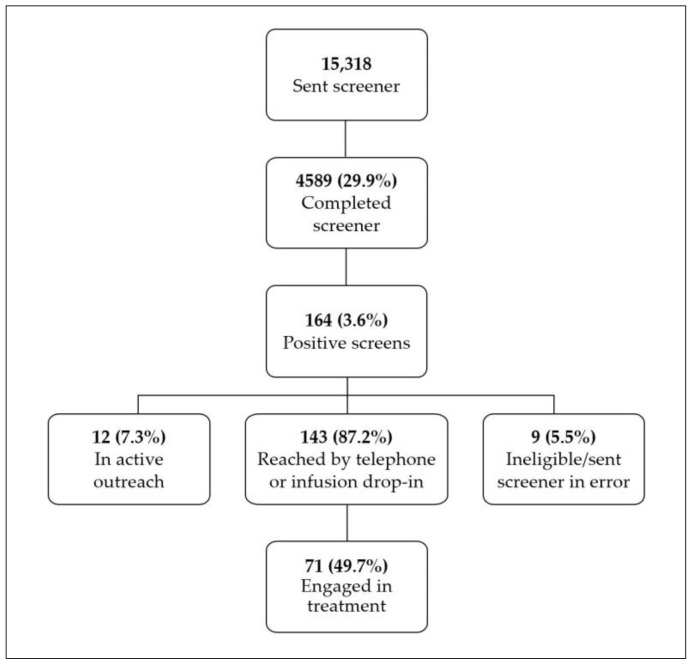
Tobacco screener completion and referrals for unique patients between 25 June 2019 and 6 April 2020. Smoking within the last 30 days resulted in a positive screen.

**Table 1 ijerph-17-05034-t001:** Characteristics of all documented smokers in the Lurie Cancer Center patient population and of the subgroups of smokers referred and engaged in treatment.

Characteristic	Engaged (*n* = 71) *n* (%)	Total Referred (*n* = 164) *n* (%)	Total Estimated Smoking Population (*n* = 1529) *n* (%)
Sex (% female)	36 (50.7%)	83 (50.6%)	845 (55.3%)
Race (% minority)	9 (12.7%)	26 (15.9%)	348 (22.8%)
Ethnicity (% Hispanic or Latino)	8 (11.3%)	10 (6.1%)	95 (6.2%)
Primary Insurance (% Medicaid)	7 (9.9%)	13 (7.9%)	218 (14.3%)
Age (mean, years)	59.6	58.8	58.3

Notes. Health insurance information includes Medicaid/Medicaid replacement and reflects insurance status at the end of the 9-month observation period. Smoking prevalence data are based on established patients with at least one visit during the observation period and may be limited by inconsistent documentation (only 82% of patients were assessed).

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
