# Peer review of "Leveraging Patient Reported Outcomes Measurement via the Electronic Health Record to Connect Patients with Cancer to Smoking Cessation Treatment"

_ijerph, 2020, doi:10.3390/ijerph17145034_

Round 1
Reviewer 1 Report
1 Repeated surveys of oncology providers demonstrate that most ask and advise, Few discuss medications or actively provide smoking cessation support (J Thorac Oncol. 2015;10: 1532–1537). As the authors agree, this discussion is of therapeutic benefit. That process should not be sidelined in any way. Oncologists fall down with discussing and providing treatment. In recommending PRO as a promising solution for common barriers to consistently assess tobacco use is overstated. They did not show that the PRO identified more smokers, or a select goup of smokers, than did routine care They did not tell us the number of smokers identified by routine care. They did not tell us if they identified smokers that had been missed by usual care.
Linking a smoking diagnosis to a referral is a separate and needed practice. and not related to the use of the PRO.
- Whether using the PRO questionnaire captures additional patients for referral is not sufficiently supported in this paper.
- a) We do not know if the providers were aware that the PRO linked tool automatically led to referral, If they did, then their own referral rate is biased by knowing an automatic referral process was in place.
- b) We do not know how many of the patients who filled the questionnaire had already been identified as smokers and if the questionnaire identified any additional patients . We have no data to tell us that this questionnaire reached an otherwise missed population, so we cannot conclude that the stigma of smoking was reduced,
- One cannot say that engagement with treatment was improved because only 11% of the estimated number of smokers replied, and those who replied were a select subgroup.
Any conclusions on rate of participation in treatment is biased by selecting for a population who voluntarily replied about their smoking status
- Making the questionairre " available" as opposed to required, or as opposed to answering a direct question from the provider, undermines the critical nature of knowing the smoking status of the patient. Including smoking status in a questionnaire along with dietary issues and insurance issues, also belies the vital significance of the response. Current practice supports a much stronger approach i.e Opt Out. The authors recognize that this tool should not replace current screening practices, however they have not shown what it adds.
- As is discussed in the paper, hiding ones smoking status, especially in the "recently quit" subgroup is not uncommon. However when only 28.5% of the patients answered the questionnaire, very many smokers may be hiding among the non-responders. Of the 14,732 patients who received the screener only 28.5% responded. 164 were identified as smokers or 1%. The analytics report is quoted as estimating 1517 smoker in the population in that time frame, thus the questionnaire may have identified 11% of smokers, or missed 89%
- We do not know the demographics of the population, and cannot judge if there is a racial or socioeconomic bias.
5..The assumption that smoking cessation is too time consuming for an oncologist to address should be questioned. We have no computerized system that successfully replaces the simple role of the oncologist to ask the patient about their smoking and to strongly advise them to proceed with suggested treatments.
Author Response
Reviewer 1
Point 1: PRO measurement as a promising solution for common barriers to the assessment of tobacco use in cancer patients is overstated. Most oncology providers ask about tobacco use and advise their patients to quit smoking. Treatment referral is a separate and needed practice and not related to the use of the PRO system.
Response: Our view of the role of PRO measurement is that it should serve as a supplement to clinic-based identification of tobacco use to support the cancer care team and is a role strongly preferred by oncology providers (Day et al. 2018; page 2, lines 78-80). Throughout the revised manuscript, we have clarified our view of the supporting role of the PRO tobacco screening and automated referral system in the goal to identify tobacco use in every patient who smokes and connect them with treatment. We now reference on page 8 (lines 237-239) the study by Warren et al. (2015) which found that many oncologists ask about smoking and advise their patients to quit, but few actively treat or refer for treatment. We agree that connecting patients to treatment is a second step, which we believe is facilitated by the PRO system because a positive screen results in an automated message to the program and then proactive outreach to the patient.
Papers cited:
Day F.L., Sherwood E., Chen T.Y., et al. (2018). Oncologist provision of smoking cessation support: A national survey of Australian medical and radiation oncologists. Asia-Pacific Journal of Clinical Oncology, 14, 431‐438.
Warren, G.W., Dibaj, S., Hutson, A., Cummings, K.M., Dresler, C., & Marshall, J.R. (2015). Identifying targeted strategies to improve smoking cessation support for cancer patients. Journal of Thoracic Oncology, 10, 1532-1537.
Point 2: Whether PRO measurement captures additional patients is not adequately supported. To what extent did the PRO system identify more smokers or a different subgroup of smokers, as compared to smoker identification and referral through routine care?
Response: As described in the revised manuscript (page 8, lines 247-248), only 6 patients (3%) were referred through both pathways. Though preliminary given the small sample sizes, a greater proportion of the patients reached through the provider pathway were female, member of a racial minority group, and insured through Medicaid. A priority of the ongoing refinement of the PRO system will be to identify strategies to better reach the patients who are in greatest need of tobacco cessation treatment (page 9, lines 285-289).
Point 3: It is unclear whether providers were aware that the PRO system automatically led to a referral. Knowledge of this procedure would have biased the provider referral rate.
Response: We agree and describe this possibility on page 8 (lines 248-251). This observation highlights the importance of communicating the automated PRO referral to the oncology team as soon as it is received and providing regular updates on engagement and treatment outcomes (page 9-10, lines 319-325). In the revised manuscript (page 5, lines 157-161), we have added details about the training sessions conducted with providers and staff on the ways in which they can refer patients to our tobacco treatment program. Providers’ knowledge of the PRO/MyChart tobacco screening and automated referral system and their reliance on it to help support tobacco dependence care should increase with repeated training.
Point 4: The authors should be careful about concluding that the PRO procedure reduced the stigma of smoking as compared to the provider context.
Response: We agree and now present this as a possibility (page 9, lines 274-279).
“The initial success of PRO measurement may be due in part through reducing the stigma that many smokers report experiencing following their diagnosis of cancer, especially smoking-attributable cancer (Chapple, Ziebland, & McPherson, 2004). This experience may lead to under-reporting of smoking in the face-to-face context with providers (Curry, Richardson, Xiao, & Niaura, 2012). PRO tobacco screening may lessen the feeling of stigma through enabling patients to report on their smoking status electronically outside of the clinic setting.”
Point 5: One cannot say that engagement with treatment was improved because only 11% of the estimated number of smokers replied, and those who replied were a select subgroup. Any conclusions on rate of participation in treatment is biased by selecting for a population who voluntarily replied about their smoking status.
Response: We have edited the manuscript to be more careful about the conclusions we draw about the effectiveness of the PRO system. As stated in the revised manuscript (page 9, lines 296-301), our experience with PRO measurement thus far is that this strategy has the potential to play an important role as a part of a multi-component tobacco screening and treatment engagement approach. In addition, we now briefly comment on our ongoing efforts to improve the PRO component through increasing the rates of PRO screener completion and treatment engagement among smokers (page 9, lines 308-312).
Point 6: Making the screener "available" as opposed to required, or as opposed to answering a direct question from the provider, undermines the critical nature of knowing the smoking status of the patient. Current practice supports a much stronger approach, i.e., opt out. The authors recognize that the PRO measurement should not replace current screening practices; however, they have not shown what it adds.
Response: We agree on the critical importance of assessing/documenting tobacco use status in every cancer patient at every oncology visit and have clarified this in the revised manuscript (page 2, lines 46-52). We believe that the PRO system is consistent with an opt-out approach (e.g., Richter & Ellerbeck, 2015) in that the screener is automatically sent to all cancer patients on a monthly basis. Patients who screen positive for current or recent smoking are automatically referred for treatment regardless of readiness to quit or interest in treatment. In addition, we advise providers and other members of the oncology team to update smoking status at every visit. Monthly completion by every patient would enable providers to spend more time encouraging smoking cessation and treatment engagement and utilization. Our preliminary results suggest that the PRO system may contribute to high patient satisfaction with cancer treatment. Another issue that we plan to examine in future research is the extent to which providers feel that the PRO system adds value. Other potential advantages, which warrant further study, are briefly described on page 9 (lines 280-289) and page 10 (lines 326-327).
Paper cited: Richter, K.P., & Ellerbeck, E.F. (2015). It's time to change the default for tobacco treatment. Addiction, 110, 381-386.
Point 7: The PRO only reaches a small proportion of the population of smokers (about 11%).
Response: We agree that this is a limitation. Increasing the completion rate is an important current focus, which we are attempting to achieve in part through administering the tobacco use screener in the clinic prior to the appointment for patients who do not complete it beforehand. This limitation highlights the importance of our multicomponent approach to reaching patients: 1) PRO tobacco use screening prior to oncology visits and automated referral; 2) provider referral of smokers seen in oncology clinics; and 3) a cancer patient tobacco use registry to identify and proactively reach smokers who are not receiving treatment. The limitation of low response rate and our approach are mentioned in the Discussion (page 9, lines 306-318).
Point 8: The demographics of the population are unknown so one cannot judge if there is racial or socioeconomic bias in the subgroup of patients reached by the PRO system.
Response: In the revised manuscript (page 9, lines 282-285 and lines 306-308), we note this as a potential issue that needs further examination. We also added the demographics of the entire referred and smoking population (Table 1). We also have added a brief discussion (page 9, lines 285-289) of strategies being considered to increase the reach of the PRO system to racial and ethnic minorities and patients of lower socioeconomic status: 1) telephone-based tobacco use screening for patients who are not enrolled in MyChart; 2) Geo-fenced targeted advertising campaigns for specific underrepresented zip codes; and 3) in-clinic completion of the PRO tobacco screener.
Point 9: The assumption that addressing smoking cessation is too time consuming for an oncologist should be questioned. We have no computerized system that replaces the simple role of the oncologist to ask the patient about their smoking and to strongly advise them to proceed with suggested treatments.
Response: Though there are studies that support this perception among some oncologists, we acknowledge that there is a lack of recent literature on the topic. In the revised manuscript, we also acknowledge that lack of time may not be a barrier for many oncologists and that the level of concern likely varies by healthcare system and provider level factors (page 2, lines 64-69, 71-73). We also refer to the critical role that oncologists have in addressing tobacco use and dependence throughout the revised manuscript.
Reviewer 2 Report
Dear authors,
I have read and I have found your study very interesting. Nevertheless, I would be happy if you could reconstruct the abstract. Abstract is written in a way that it gives the impression that the study there is nothing to add to the subject of smoking. But, going through the whole manuscript I realized the importance of the paper. Purpose of the study is missing, so it should be added. Please, state what kind of a study is this and do not forget to mention the ethical approval. Although I understand the statistical analysis is descriptive, on the other hand it is very simple. In order to ensure and obtain safe results, statistical analysis should be rather strict. References are extensive and relevant.
Author Response
Reviewer 2
Point 1: The abstract needs to include the purpose and importance of the study.
Response: We have added the purpose of the study and its potential contribution to the literature (page 1, lines 23-26, 30-36).
Point 2: Describe the type of study and note the ethics approval.
Response: We utilized a descriptive program evaluation study design with an approximate 9-month observation period. We have added the ethical approval by the Northwestern University Institutional Review Board (STU00210245). Please see page 3 (lines 99-102).
Point 3: Given the lack of statistical analysis of outcomes, the authors should be cautious about drawing conclusions regarding differences observed in the outcomes of interest.
Response: We have revised statements throughout the manuscript to be more cautious in our interpretations of the significance of changes in the PRO reach and engagement outcomes and the observed differences in outcomes between the PRO and provider referral strategies. To convey methodological rigor, we have added a brief description of the data extraction and validation process involving the Northwestern University Enterprise Data Warehouse and Northwestern Memorial Hospital Analytics Department (page 6, lines 173-179).
Reviewer 3 Report
Title:
Leveraging PROs to Connect Patients with Cancer to Smoking Cessation Treatment
Overall comment:
- Well-written.
- Could include patients from other health systems or the duration of the study could be longer to ensure reliability and validity of the study.
Abstract:
- Well-written.
Introduction:
- Well-written.
- Could include some data regarding tobacco related cancer mortality/morbidity to accentuate the problem.
- Overall background is good, and the problem is nicely described. What is missing in the introduction is how did the researchers identify gap in the literature? Has this area previously been explored?
Methods:
- Well-written.
Results
- Well-written.
Discussion:
- Well-written.
- Nicely discussed advantages of the PRO method and how it helps in improving public health.
Minor correction:
- Line 141, “…April 6th, 2020..”
- Line 187, “…experiencing stigma…”
- Line 205, “…types. Having….”
Author Response
Reviewer 3
Point 1: The longer observation period would ensure reliability and validity of the outcomes.
Response: Given the limited observation period (9 months), we agree that the results are preliminary and require longer-term evaluation and formal statistical testing before conclusions can be drawn regarding the effectiveness of the PRO system. That recognition is what led us to submit the manuscript as a Short Communication. We added this limitation to the Discussion (page 10, lines 340-343), and have been careful throughout the revised manuscript to characterize the results as preliminary.
Point 2: The Introduction should include data on tobacco-related cancer mortality/morbidity and the researchers’ motivations that drove the development of the PRO system.
Response: We now describe the high significance of cigarette smoking and tobacco-related cancer mortality and morbidity (page 2, lines 41-44). We also briefly describe the gaps in care at Northwestern Medicine that motivated us to develop the PRO tobacco screening and automated referral system. Briefly, the system was designed to address insufficient tobacco screening and treatment referral. PRO measurement had been successfully applied to other under-addressed clinical problems, such psychological distress screening, among our cancer patient population and was expected to help address tobacco use (page 2, lines 58-60).
Point 3: Minor corrections are needed: Line 141, “…April 6th, 2020..” Line 187, “…experiencing stigma…” Line 205, “…types. Having….”
Response: We have made these corrections, and carefully reviewed the manuscript for additional typos and grammatical and punctuation errors.
Reviewer 4 Report
This manuscript reports on a treatment recruitment strategy for smokers who are engaged in cancer treatment care, utilizing electronic health records to facilitate smokers self-identifying as currently smoking and then connecting them with treatment. The authors are defining the tobacco screener they used as a patient reported outcome. A large proportion of cancer patients smoke, and it is important but also difficult to both identify them and then connect them with proper smoking cessation treatment. This manuscript is of importance but has several limitations that limits its interpretation and impact.
Major Comments
- The authors provide demographic data on participants who engaged in treatment, but it would also be valuable to see this kind of information for those who did not participate in treatment. Further, information on what type of pharmacotherapy people received, who completed all of the tobacco treatment and what treatment options they selected, and what their quit status was would be helpful.
- Lines 161-162: The numbers provided do not seem to be 100% of the participants who answered the satisfaction survey. In Line 157 the numbers reported are different.
- This PRO seems to be basically a tobacco use screener. Some sort of discussion about how these stand-alone tobacco questions, versus tobacco questions integrated within a questionnaire that had other PRO, would be useful. Additionally, defining a PRO for this reading audience, and why they are defining this tobacco screener as a PRO, would be useful.
- It would be helpful if the authors provided more information on participants who did not complete the PRO tobacco screener.
- The authors report that only 4% of people who completed the screener were current smokers, which seems quite low for a cancer population. The authors should address this in their discussion section. In the discussion section the authors should also address limitations considering 72% of people did not complete the screener.
- What is the total smoking population? Do they give us those numbers? In Line 176-177 they talk about percentages of the total smoking population, but it is not clear where they told us what the overall prevalence of smoking was. Theoretically this information should be available in the EHR.
- There are a number of groups working to increase referral of smokers both in cancer settings and also utilizing EHRs. Integrating the results of those studies in the discussion (both similarities and contradictions in results) is helpful for interpretation.
Minor Comments
- I would recommend not including the acronym PRO in your title, as all readers may not know what PRO stands for.
- Were there any patients both captured by the PRO screening tool and referred by their provider?
- Having a flow chart showing the numbers who were screened and participated at each step would be helpful for the reader.
- Figure 5 and 6 should be one figure.
- Some small editing issues (example: Line 214, omission of “to” – should be “has been shown to lead to poorer satisfaction…”).
Author Response
Reviewer 4
Point 1: It would also be valuable to see the demographics of those who did not participate in treatment, information on what type of pharmacotherapy people received, who completed all of the tobacco treatment, what treatment options they selected, and their quit status.
Response: As requested, we have added the demographic characteristics of the sample of patients who screened positive for current smoking and were referred for treatment (page 7, table 1). In addition, we have added available data on treatment utilization (page 7, lines 197-202). Data on smoking cessation is not yet available due to observation period, but the authors agree this would be a valuable addition to a future manuscript on the PRO system.
Point 2: Lines 161-162: The numbers provided do not seem to be 100% of the participants who answered the satisfaction survey. On line 157 the numbers reported are different.
Response: We have clarified the reason for the noted discrepancies (page 7, lines 218-220). Twenty-seven patients answered some or all of the satisfaction survey items. Several patients did not respond to certain questions or chose not to complete the entire survey.
Point 3: This PRO seems to be a tobacco use screener. Some sort of discussion about these stand-alone tobacco screening questions, versus tobacco screening questions integrated within a questionnaire along other PRO measures, would be useful.
Response: As now described on page 3 (lines 111-114), patients receive a bundled set of PRO measures, which also include assessments of pain, physical function, fatigue, depression, anxiety, and other potential adverse effects of cancer treatment. All measures are assigned together (i.e., bundled), but due to EHR system functionality some of them appear as separate questionnaires in the patient view (see Figure 2). The other key aspect of the system is the automated referral that is triggered by a positive screen.
Point 4: Provide a definition of PRO and justification for the tobacco screener as a PRO.
Response: We have improved the definition of PRO (Cella et al, 2015) and the justification for the current application to tobacco use (page 2, lines 56-58, 64-69): reduced burden on cancer care team to conduct the screening, consistent administration (i.e., assignment via MyChart), ability to offer treatment to patients who previously declined treatment, and the potential to reduce stigma. As discussed on page 2 (lines 58-60), screening is a common goal for PRO measurement (Austin et al, 2020).
Papers cited:
Cella D, Hahn EA, Jensen SE, et al. (2015). Patient-Reported Outcomes in Performance Measurement. RTI Press: Research Triangle Park (NC).
Austin, E., LeRouge, C., Hartzler, A.L. et al. (2019). Capturing the patient voice: Implementing patient-reported outcomes across the health system. Quality of Life Research, 29, 347-355.
Point 5: It would be helpful if the authors provided more information on participants who did not complete the PRO tobacco screener.
Response: We agree that this information would be valuable. At the present time, the system does not have the capability to characterize non-completers. In the revised manuscript, we describe this as a target for system enhancement (page 9, lines 306-308).
Point 6: The authors report that only 4% of people who completed the screener were current smokers, which seems quite low for a cancer population. The authors should address this in their discussion section. In the discussion section, the authors should also address limitations considering 72% of people did not complete the screener.
Response: We agree that the low response rate is a current limitation, which was also noted by Reviewer 1. Increasing the completion rate is an important focus, which we are attempting to do through administering the tobacco use screener in the clinic prior to the appointment for patients who do not complete it beforehand (page 9, lines 308-312). We agree that the estimated smoking prevalence of 4% seems low. However, this relatively low prevalence is comparable to other academic healthcare population-based studies (Jenssen et al, 2019, Gali et al, 2020). The low rate overall may be driven in part by the sociodemographic characteristics of the Lurie Cancer Center population and some underreporting and relatively low documentation rate. We estimate that only 70-80% of patients were assessed/documented during the past 6 months.
Papers cited:
Jenssen, B.P., Leone, F., Evers-Casey, et al. (2019). Building systems to address tobacco use in oncology: Early benefits and opportunities from the Cancer Center Cessation Initiative. Journal of the National Comprehensive Cancer Network, 17, 638-643.
Gali, K., Pike, B., Kendra, M.S., et al. (2020). Integration of tobacco treatment services into cancer care at Stanford. International Journal of Environmental Research and Public Health,
17, 2101.
Point 7: It would be helpful to know the prevalence of smoking among the patient population.
Response: The estimated prevalence of smoking among the approximately 82% of patients with updated smoking status during the observation period was 4.2%. We have added this information to the revised manuscript (page 6, lines 188-190).
Point 8: There are a number of groups working to increase referral of smokers both in cancer settings and also utilizing EHRs. Integrating the results of those studies in the discussion (both similarities and contradictions in results) is helpful for interpretation.
Response: We have added a brief discussion of the other cancer centers that have developed and evaluated population-based tobacco cessation programs to optimize reach (page 8, lines 239-243).
Paper cited: D'Angelo, H., Rolland, B., Adsit, R., et al. (2019). Tobacco treatment program implementation at NCI cancer centers: Progress of the NCI Cancer Moonshot-funded Cancer Center Cessation Initiative. Cancer Prevention Research, 12, 735-740.
Point 9: Do not include the acronym PRO in the title.
Response: We revised the running head as requested. PRO was not used in the title.
Point 10: Were patients both captured by the PRO screening tool and referred by their provider?
Response: Only 6 patients were both captured by the PRO screening tool and referred by their provider. This observation has been added to the manuscript (page 8, 247-248).
Point 11: Having a flow chart showing the numbers who were screened and participated at each step would be helpful for the reader.
Response: As suggested, we have added a flow chart (page 6, figure 5).
Point 12: The data in Figure 5 and 6 should be integrated into a single table.
Response: Table 1 now integrates the demographic and clinical characteristics of the engaged sample.
Point 13: Some small editing issues (example: Line 214, omission of “to” – should be “has been shown to lead to poorer satisfaction…”).
Response: We have made these corrections, and carefully reviewed the manuscript for additional typos and grammatical and punctuation errors.
Round 2
Reviewer 4 Report
The authors have significantly improved the manuscript. My only minor comments are in regards to their statement about tobacco screening and treatment referrals via EHR not yet being studied (I question whether this is true), and also whether or not they should include more results from other C3I centers who are conducting research.
Author Response
Reviewer 4
Point 1: Questionable accuracy of the statement about novelty of EHR-based tobacco use screening and treatment referral.
Response to Point 1: We agree that EHR-based tobacco use screening and referral has been well studied. We have clarified on page 2 (lines 62-63) and throughout the manuscript that, to the best of our knowledge, automated tobacco use screening and referral via the EHR patient portal is the novel aspect of this study.
Point 2: Consider including results from other C3I centers.
Response to Point 2: As suggested, we now cite additional 3CI-related publications and have added brief discussion of their contributions (page 8, lines 237-245).
Additional papers cited:
- Ramsey, A. T., Chiu, A., Baker, T., Smock, N., Chen, J., Lester, T., . . . Chen, L. S. (2019). Care-paradigm shift promoting smoking cessation treatment among cancer center patients via a low-burden strategy, electronic health record-enabled evidence-based smoking cessation treatment. Translational Behavioral Medicine. Online ahead of print.
- Meyer, C., Mitra, S., Ruebush, E., Sisler, L., Wang, K., & Goldstein, A. O. (2020). A lean
quality improvement initiative to enhance tobacco use treatment in a cancer hospital.
International Journal of Environmental Research and Public Health, 17(6), 2165.
- Sheffer, C. E., Stein, J. S., Petrucci, C., Mahoney, M. C., Johnson, S., Giesie, P., . . . Hyland,
- (2020). Tobacco dependence treatment in oncology: Initial patient clinical characteristics and
outcomes from Roswell Park Comprehensive Cancer Center. International Journal of
Environmental Research and Public Health, 17(11).
- D'Angelo, H., Ramsey, A. T., Rolland, B., Chen, L. S., Bernstein, S. L., Fucito, L. M., . . .
Baker, T. B. (2020). Pragmatic application of the RE-AIM framework to evaluate the
implementation of tobacco cessation programs within NCI-designated cancer centers. Frontiers
in Public Health, 8, 221.
